# The possibility of sports industry business model innovation based on blockchain technology: Evaluation of the innovation efficiency of listed sports companies

**Chenchen Lv**[1,2]*, **Yifeng Wang**[1‡], **Chai Jin**[2‡]

**1** School of Economics & Management, XIDIAN University, Xi'an, Shaanxi, China, **2** School of Sports Economics and Management, Xi'an Physical Education University, Xi'an, Shaanxi, China

‡ These authors also contributed equally to this work.
* lvchenchen24@stu.xidian.edu.cn

## Abstract

The combination of blockchain and the sports industry is bound to be a trend in terms of innovation and the development of sports industry in the near future. Therefore, after analyzing the problems existing in the current business model of the sports industry, this paper takes 200 innovation indicators of 50 listed companies in the sports industry, among which 15 use blockchain and 35 do not use, as research samples and uses a three-stage DEA model to calculate their innovation efficiency. The results show that sports companies that use blockchain have better performance in terms of innovation efficiency than do those that do not, which illustrates the necessity of blockchain-based business model innovation. Then, by focusing on the internal structure and the case of the "*vSport* blockchain", this work shows that this business model innovation is feasible. The article first proposes a blockchain-based sports industry business model and then uses a combination of empirical and case studies to prove the necessity and feasibility of sports industry business model innovation.

## Introduction

Since 2017, blockchain has gradually attracted the attention of both the public and the capital market. With the sweeping emergence of blockchain technology, Internet giants, such as Tencent, Alibaba, and Baidu, have experimented with blockchain, and blockchain companies have sprung up all over the world. At present, many countries around the world have been promoting blockchain technology. Moreover, the sports industry has had the fastest growth rate after humankind entered the second decade of the new century. According to PR Newswire, the market value of the global sports industry is growing at a compounded annual growth rate of 8.1% per year and will reach US$253.465 billion by 2024. The emergence of blockchain has brought about a new dawn for the growth of the sports industry. The sports industry will once again usher in a revolutionary improvement with the advent of blockchain technology, which seems to be agreed upon by those actors in the industry.

**Data Availability Statement:** The data of "degree of regional technological development" cannot be shared publicly because of the confidentiality agreement, but others can access these datasets

through the official website of World intellectual Property Organization or contact this email address (kyle.bergquist@wipo.int) to apply for data access. We filed an application to the World Intellectual Property Organization and obtained the data, and we did not have any special access privileges that others would not have.

**Funding:** The authors received no specific funding for this work.

**Competing interests:** The authors have declared that no competing interests exist.

Therefore, in this important historical process, how to effectively study and learn the essence of "blockchain + sports", how to clarify and focus on the application fields and implementation scenarios of blockchain technology in the sports field, and how to explore the innovation of sports blockchain business models are the key factors that decide whether the Chinese sports industry can stand at the forefront of global sports competition in the future. From a theoretical point of view, the existing research on sports blockchain is still in its infancy. Different from previous summary studies, this article discusses "blockchain + sports" from the perspective of business model innovation, which will enrich the research results in this field. From a practical point of view, the combination of blockchain and the sports industry is bound to become a trend in terms of innovation and development in the future. At the same time, there are some shortcomings in the current sports industry business model, which limit the excitation of market potential and the development of the entire industry. Given the openness, authenticity, and decentralization of blockchain, a new business model of the sports industry is formed, which is more suitable for future sports market development. Thus, the research in this article provides theoretical guidance on the combination of the blockchain and the sports industry, as well as its feasible business model innovation.

Therefore, this article first explains the existing problems of the current sports industry business model. Then, the necessity of the sports industry business model innovation will be illustrated through a three-stage DEA efficiency evaluation, which takes 50 listed companies involved in the sports industry as research samples and compares the innovation efficiency between companies that use blockchain concepts or technologies and those that do not. Then, by focusing on the internal logic and structure of the business model innovation of "blockchain + sports", as well as the longitudinal study on the case of "vSport blockchain", the feasibility of this business model innovation is proven.

## Related studies

### Blockchain

Formal research on blockchain began in 2015. The initial research on blockchain focused on computer software and its application in the financial field. He Zhao proposed a kind of authenticity guarantee method of sensor data based on blockchain technology in the field of encrypted digital currency [1]. Swan explained that blockchain is essentially a public ledger, which can be a global, decentralized record for registration, inventory and transfer. It records all assets, not only finances but also property and intangible assets such as votes, software, health data and ideas [2]. Since then, academic circles have continuously expanded the research on blockchain technology [3]. In terms of basic research, Liu et al. [4] designed a distributed trusted network connection architecture based on blockchain; at the same time, the main advantages of blockchain technology, such as decentralization, high-efficiency mechanisms, and process security, have led to extensive discussions among experts and scholars in the fields of logistics and finance [5,6]. Azaria et al. [7] focused on how to use blockchain for medical data access and authority management. Deng and Li [8] discussed the application of blockchain smart contract technology in the supply chain factoring business through game theory. While blockchain technology may bring about more reliable and convenient services, Lin and Liao [5] raised various issues concerning blockchain technology, such as technical issues, the applied range of blockchain, comprehensive costs and regulations.

### Characteristics of blockchain

Blockchain is a kind of chained data structure in which data blocks are sequentially connected in chronological order, and it is a distributed ledger that cannot be tampered with and is

guaranteed by cryptography [9]. Simply put, blockchain technology refers to a way for all people to participate in bookkeeping. In blockchain, each node can maintain accounts, and the system judges the fastest and best node during this period of time and records its content on the ledger. At the same time, the system sends the content of the ledger to other nodes in the system for back up, so each node in the system has a complete ledger [10]. This method, which we call blockchain technology, is also the origin of blockchain decentralization. This method can ensure the security of the ledger. Through the collation of existing studies, the characteristics of blockchain are summarized below.

First, there is no central ledger, so it cannot be destroyed [9]. In the past, behind the accounting system, there was always a pair of "large hands" that controlled all information in the background, and the inflow and outflow of information had to go through these "large hands", which allowed for "turning hands into clouds and covering hands as rain". However, each node in the blockchain is a part of the system, and each has equal rights and exactly the same ledger, so destroying part of the nodes has no effect on the system as a whole [11,12].

Second, blockchain technology guarantees the authenticity of the ledger. Because it cannot be tampered with, unless you can control and modify the computers of most people in the system, the system will refer to the opinions of most people to determine the real result [13–15]. In addition, because others do not admit it, then there is no point in modifying one's own ledger. Blockchain technology can improve public record management in many ways. When users put content on the chain, it can automatically prove its authenticity [16].

Third, since there is no centralized intermediary organization, everything is automatically run through preset procedures, which not only tremendously reduce transaction costs but also improve transaction efficiency [17]. The advantages of a centralized system can only be exhibited within the system itself. Once other systems are involved, the efficiency cannot be improved [18]. However, as long as they are on the same blockchain, different institutions only need smart contracts to realize the linkage between value and equity. Moreover, since everyone has the same ledger, the openness and transparency of the ledger recording process is ensured [11,19,20].

Generally, blockchain research has become more extensive and in-depth. Scholars have performed many studies on the technology, definition, characteristics, and frameworks of the blockchain as well as its integration with computer technology, finance, logistics, public management, corporate management and other fields, but the research on blockchain is still in its infancy, as it is mostly theoretical and predictive. As the application of blockchain has become increasingly extensive, the related research has been enriched empirically and intensified.

## Business model and business model innovation

Regarding a business model, there are many misunderstandings and deviations among academic and business circles. DaSilva and Trkman [21] combined a large amount of the literature in 2014 and found that many people confused business models with strategies. Amit and Zott [22] pointed out that a business model is designed to create value through the use of business opportunities, thereby designing transaction content, transaction structure and transaction governance. Zott and Amit [23] also pointed out that a business model clarifies how an organization is connected with external stakeholders and how it conducts economic exchanges with these stakeholders to create value for its trading partners.

Research on business model innovation started in 1998. The relevant research is relatively sufficient, though still increasing, and there remain many fields that can be explored. The existing research is mainly divided into two directions: one focuses on the general research of business model innovation, and the other focuses on integration with various industries, such as the medical, e-commerce, and construction industries.

**Table 1. Different research perspectives on business model innovation.**

| Research perspectives | Representative | The main points |
|---|---|---|
| Corporate operation perspective | Malhotra, Staehler, Gary Hamel, Michael Hammer, Osterwalder | The business model comes from the daily operation and management of the enterprise. Therefore, the innovation of the business model can be carried out through all aspects of the company's operations [24]. |
| Strategic planning perspective | Wolfle, Knecht, Mitchell, Weill | To some extent, the business model represents the overall strategy of an enterprise. Therefore, they advocate the use of methods that are similar to strategic planning to study enterprise business model innovation. |
| Value system perspective | Christensen, Magretta, Gordijn, Hacklin | Emphasizing the positive impact on value creation from business model innovation, they believe that value changes are the primary driving force for business model innovation [25]. |
| Module reorganization perspective | P·Weill, Gordijn, Osterwalder, Johnson, Lindgardt | The business model is composed of different modules, so business model innovation needs to start with each module and the interconnection relationship between these modules [26]. |
| Element composition perspective | Zott&Amit, Di Xu, Zhilong Tian, Tao Zeng, Wei Wei, Wuxiang Zhu | These studies focus on the components of the business model and advocates the innovation of elements to achieve the innovation of the overall business model. For example, Zott and Amit summarized the attributes of business model innovation from four levels—efficiency, complementarity, lock-in function and novelty [27]—and then analyzed how to innovate the elements and the connection between them to finally achieve business model innovation. |

The existing research has mainly studied business model innovation from five perspectives: the corporate operation perspective, strategic planning perspective, value system perspective, module reorganization perspective, and element composition perspective; see Table 1.

Generally, scholars mainly conduct research from different perspectives of business model innovation, such as the value chain perspective, overall perspective, and element perspective. At the same time, researchers have a deeper understanding of the motivation, approach, and implementation of business model innovation while penetrating the field.

As far as "blockchain + business model innovation" is concerned, there are few existing studies, which mostly focus on theoretical and predictive qualitative research. For example, Morkunas et al. [28] explained the terminologies and working principles of blockchain transactions and how private and public chains affect business models. Zhang and Zhu [29] analyzed three different deployment methods based on blockchain: the private chain, alliance chain, and public chain, as well as the main application scenarios of business model innovation.

On the whole, the research on enterprise business model innovation based on blockchain technology and concepts is still in its initial stage. The related literature is limited, and most studies are theoretical and conceptual descriptions of "blockchain + business model innovation". In addition, case analysis is the main research method used in these studies.

## Blockchain + business model innovation + sports industry

**Business model innovation of sports companies.** There are relatively few studies on the business model innovation of sports companies, and most of them are related to specific sports or subindustries. For example, McNamara et al. [30] tried to empirically test whether there is more than one stable business model in the British Premier League football industry. Moreover, Aversa et al. [31] conducted a qualitative comparative analysis of the companies participating in Formula One and found the following: "There are two business models related to high performance—one focuses on selling technology to competitors, and the other focuses on the development and trading of human resources with competitors". Finally, Liu et al. [32] analyzed the business model of China's sporting goods manufacturing industry and proposed the innovation strategy of the business model of China's sporting goods manufacturing enterprises.

The business model of the sports tourism industry is relatively detailed. For example, first, based on a comprehensive perspective, Peric [33] pointed out that the sports tourism business model includes four core elements: value proposition, key resources, key processes, and value

acquisition. Second, according to the experience of sports event participants and organizers, Perić et al. [34] conducted a three-stage empirical test and proposed business models for different outdoor sports tourism projects.

**"Blockchain + Business Model Innovation + Sports".**   In recent years, research on the combination of blockchain and the sports industry has gradually increased. Many scholars have realized that blockchain technology is an innovative framework and has the potential to change the transaction model. Kshetr and Voas [35] mentioned that "blockchain can create tamper-proof voting audit trails, and the use of blockchain enabled electronic voting (BEV), which can reduce voter fraud and increase voter access". Bernstein [36] proposed the use of smart contracts in the sports industry and the application of blockchain technology to traditional sports contracts to transform them into smart contracts to completely change the common business practices and procedures. Grow and Grow [37] proposed that the protection of trade secrets has become a hot issue in the professional sports industry because blockchain technology has excellent applicability in protecting commercial data from being leaked.

Yu [38] studied the innovative development of sports big data integration and transmission paths based on blockchain, while Zhou et al. [39] discussed the issue of blockchain technology driving the innovation and development of the sports industry and pointed out that sports blockchain has positive effects on cross-border transactions and the protection of intellectual property rights. Huang et al. [40] used the literature data method, logical analysis method, and ASMI four-step method to study the core value of blockchain technology and its application in the sports industry. Moreover, Zhang and Zhang [41] studied the application of blockchain technology for protecting sports logos.

Overall, the research on blockchain and the sports industry in other countries started earlier than that in China. In particular, some foreign companies have combined blockchain technology and sports products into the market, but in terms of research content, non-Chinese research is more focused on sports competition performance, sports betting and sports law, and there is a lack of macro research on the sports industry as a whole. Although there have been studies on the realization and application paths of the integration of the sports industry and blockchain in China, studies on the deep structure and business model of "blockchain + sports industry" are still lacking, most existing research is theoretical, and empirical analysis is rare. Therefore, this article attempts to conduct empirical research from the perspective of business model innovation to fill this theoretical gap.

## Existing business model of the sports industry

After rapid development in recent years, the structure of the sports industry in China has undergone tremendous changes and has taken shape as follows: sports events upstream of the industry, the media dissemination of sports in the middle of the industry, and some derivatives and peripheral products downstream of the industry. However, there are still many problems in the development of the sports industry and its business model in China. First, the structure of the sports industry is unreasonable; second, the economic efficiency of the industry is low; and third, the current market momentum is still relatively weak. The allocation of resources is not yet sufficient, and the current business model can stimulate market potential only to a limited extent. The existing business model of the Chinese sports industry is shown in Fig 1.

### The scale of the sports industry is still small, and the industrial structure is not reasonable enough

The industrial environment inevitably has a direct or indirect impact on the business model. First, from the perspective of industrial scale, the scale of the Chinese sports industry has been

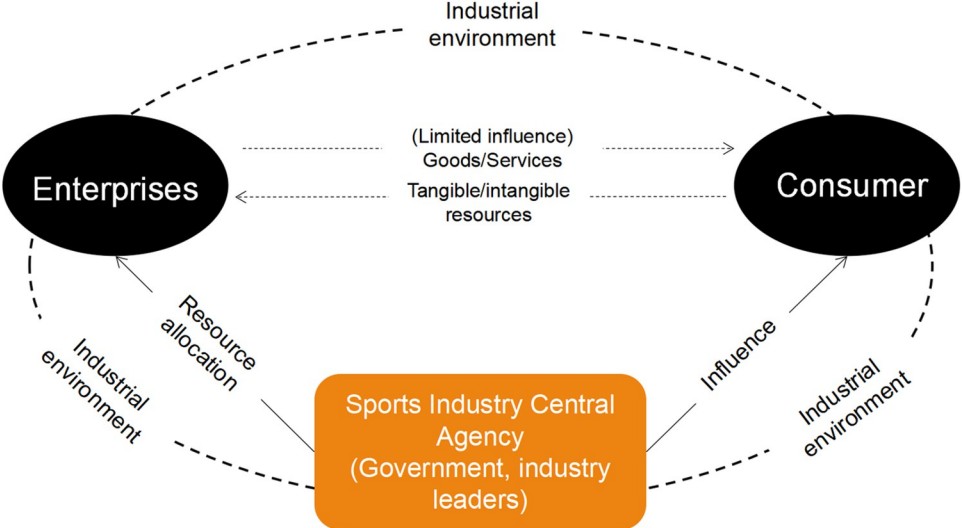

**Fig 1. Existing business model of the sports industry.**

expanding over the past 40 years. In 2019, the total scale of the sports industry reached 423 billion dollars. Compared with developed countries, there are still disparities in terms of output value and the share of GDP. For example, the output value of the sports industry in the United States in 2019 was approximately 540 billion dollars. According to the data of the China Industry Information Network in 2017, as shown in Fig 2, the sports industry's value added in South Korea, the United States, France and other countries accounted for approximately 3% of GDP. According to calculations, by 2019, the Chinese value added of the sports industry was 1,648 billion dollars, and the proportion of GDP was only 1.1%. There is still a certain gap between

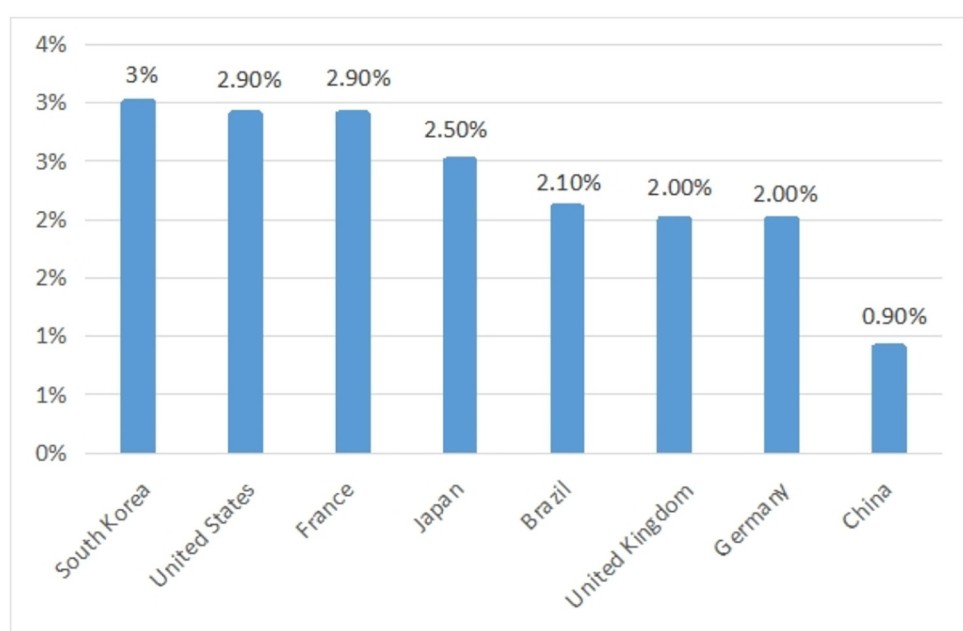

**Fig 2. The value added of the sports industry in different countries as a percentage of GDP.** Data source: China Industry Information Network.

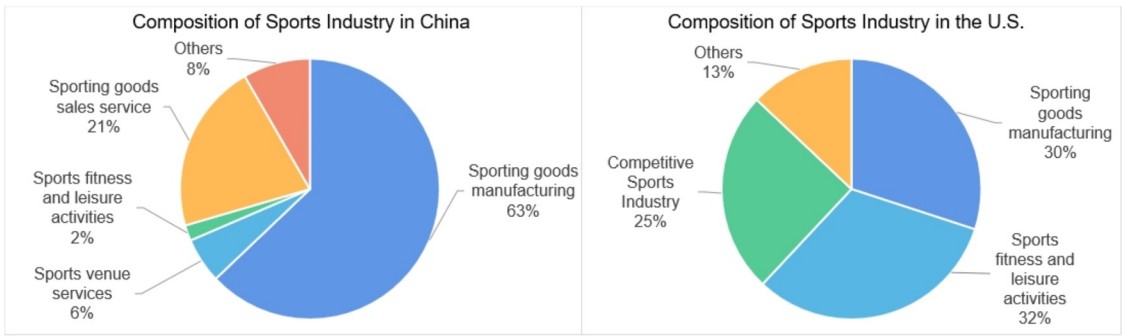

**Fig 3. Comparison of the structure of the sports industry in China and the United States.** Data source: "Gross Domestic Sport Product: The Size of the Sport Industry in the United States", Beijing Oulixin Consulting Center, National Sports General Administration.

the Chinese sports industry and developed countries in terms of industrial scale and development.

Second, in terms of the structure of the Chinese sports industry, the sporting goods manufacturing industry at the relevant level accounts for approximately 63%, which is the main supporting factor of the industry, while the sports service industry, which is the main part of the whole industry, accounts for 29%. The main reason for this is that the sports industry in China has a low degree of marketization at the early stage of its development. At the same time, the development of professional sports in China is immature, and the leisure fitness industry started later in China than in other countries and is not yet fully developed. Therefore, the output value of the sports service industry is relatively low. Compared with the mature American market, the sports service industry in China is the main supporting industry, of which fitness/entertainment, and competitive sports account for 32% and 25%, respectively. Overall, China's sports service industry only accounts for a small part of the overall structure of the sports industry, and there are still many unreasonable aspects in terms of its industrial structure configuration, as shown in Fig 3.

## Excessive centralization of sports industry resources

Since government and industry leaders have masterfully utilized most of the resources in the sports industry, the degree of marketization is not high enough, and products lack international competitiveness. Sports management systems have long faced problems such as the non-separation of government and enterprise affairs and the nonseparation of government and market affairs. Therefore, most of the important sports resources, such as sports talent, stadiums, and sports events, are in the hands of authority and industry leaders. The market did not focus enough on distribution and its the process, so the development of the entire industry has been led by the government. The government and industry leaders have mastered the "the right to control all the resources and information" in the entire industry, limiting the formation of a benign business system standard. At the same time, the so-called "relationship economy" has emerged; that is, only by having a good relationship with resource controllers can the development of the government and industry leaders be accomplished, such as being familiar with the sports bureau, club officials, and football stars. In addition, excessive centralization can limit consumers' understanding and participation in the sports industry. In addition, the largest problem exposed by the centralization of information and resources today is security. Since the centralized system controls all subnodes, once there is something wrong with the center, the entire industry faces enormous risks.

## Limited value exchange between enterprises and consumers

The value added of the sports industry mainly involves intangible assets, social benefits, economic benefits and other aspects. On the one hand, due to the particularity of the sports industry, in addition to directly providing products and services, enterprises must also use sports influence (such as sporting events IP, and concepts) to realize the value added of enterprises and consumers. However, there are still problems of information opacity, openness, and asymmetry between these two, resulting in the limited development of market potential. On the other hand, information between subindustries and related industries is relatively blocked, departmental data cannot be fully shared, and information exchange is not timely; that is, there is a "data barrier", which means that the real-time sharing of data is not realized. In addition, these "data barriers" lead to a disconnection among sports big data in the value realization process, thereby hindering the process of the coordinated development of various industry sectors. These are the reasons why the value-added effect is not nonsignificant.

Second, the current business model is unreasonable. For many sports subsectors in China, especially for some small and medium-sized enterprises, their business operating models simply copy and apply the operating models of other developed countries but lack their own characteristics and deep mining of unique resources. This mode of operation severely restricts the further development of China's small and medium-sized sports companies, thereby restricting the development of the sports industry as a whole. In addition, enterprises lack a diversified profit model. At present, most of the current profit models of sports companies rely on sponsors, such as the Chinese super league football clubs, CBA, and various leagues, which has resulted in the finding that high-value or high-quality sponsorship is even more difficult when there are more start-up companies because of the limited number of companies that can provide sponsorship.

Third, for the sports goods manufacturing industry, China mainly produces low-value-added processed and labor-intensive products. The country is seriously inadequate in terms of technology investment and technological innovation and lacks its own branded products, which has a limited brand impact on consumers. For example, many Chinese sporting goods companies started out performing contract manufacturing for international brands. However, after experiencing a rapid growth cycle, the shortcomings of the product-oriented business model and the extensive distribution and wholesale model gradually emerged. For example, after international brands such as Nike and Adidas entered the Chinese market in 2010, market competition intensified. The combined market share of these two giants accounts for more than 30% of the total market share of the Chinese sporting goods industry, and the market share of domestic companies has been declining year by year.

## Low innovation efficiency in the sports industry

PricewaterhouseCoopers recently released the "2019 PWC Sports Industry Survey Report", which pointed out that the expected growth rate of the global sports market in the next 3 to 5 years is 6.4% and that China will continue to play the role of a growth leader in this market. With technology development, digital transformation will have an impact on the development of all areas of life, and the sports industry is certainly no exception. According to the "Report", with the overall acceleration of the digital transformation trend, the pressure placed on the global traditional sports industry has doubled. According to the "Report", with the acceleration of the digital transformation trend, the pressure on the global traditional sports industry has doubled. Leaders in the sports industry have fully realized the need for innovation. Although 94% of these leaders are aware of the importance of the innovation ability of sports organizations, only 46% are currently implementing specific innovation strategies. The survey shows

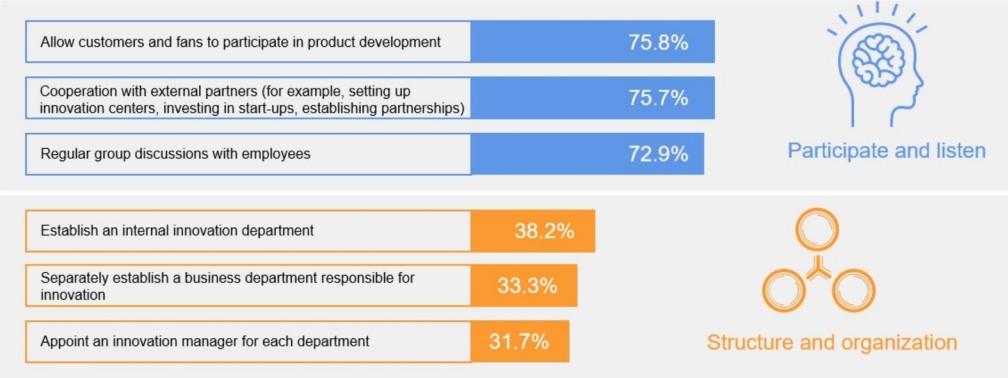

**Fig 4. The primary measures for the implementation of innovation and transformation in the traditional sports industry.** Data source: "2019 PwC Sports Industry Survey Report", N = 538.

that 75.8% of leaders in the traditional sports industry believe that allowing customers and fans to participate in product development is the primary measure for the traditional sports industry to implement innovation and transformation, as shown in Fig 4.

The current business model of the sports industry restricts innovation efficiency. On the one hand, the "centralized" business model prevents a large number of high-quality resources within the system from being released, for instance, the high-quality sports resources accumulated over the past ten years, the various training bases and stadiums all over the country, and the highest level of athletes, coaches, and referees in the country; thus, the quality and quantity of the investment in innovation are far from sufficient; on the other hand, the industry barriers formed by the existing business model of the sports industry make the innovation proceed only within the industry, thus limiting the positive impact of customers, fans and other industries, resulting in a lower conversion rate of innovation from input to output, that is, lower innovation efficiency.

The next section analyzes and compares the impact of blockchain technology on the innovation efficiency of the sports industry in detail.

## Data and methodology

In this chapter, we first analyze and comb through the 50 sports companies listed in the A-share, Hong Kong, U.S. and the New Third Board stock markets to identify companies that have introduced or used "blockchain" technology or concepts and compare them with those that have not. To compare and evaluate the innovation efficiency of innovative technology, DEA indicators regarding the aspects of acquisition and utilization, that is, input and output, respectively, are set.

### Introduction to the three-stage DEA model

In 1978, famous operations researchers A. Charnes, W.W. Cooper, and E. Rhodes first proposed the data envelopment analysis method named the C2R model, abbreviated as the DEA model, which was used to evaluate the relative effectiveness of the counterpart departments. This model was used to study the "scale efficiency" and "technical efficiency" of the "production sector" with multiple inputs and multiple outputs and achieved outstanding results at the time.

In 2002, Fried pointed out that the traditional DEA model did not consider the impact of environmental factors and random noise on the efficiency evaluation of decision-making units [42]. He successively published two articles, "Incorporating the Operating Environment Into a Nonparametric Measure of Technical Efficiency" and "Accounting for Environmental Effects and Statistical Noise in Data Envelopment Analysis", and discussed a new efficiency evaluation model, which overcomes the shortcomings of the one-stage DEA method, which cannot incorporate environmental factors and random noise for analysis. Fried et al. [42] believe that the inefficiency of enterprise production is affected not only by enterprise management but also by two exogenous factors, the environment and random error. The purpose of the three-stage DEA model is to eliminate the influence of the environment and random error to more truly reflect the efficiency of each decision-making unit.

In the first stage, we use the original input-output data for the initial efficiency evaluation. This is mainly done through the use of linear programming methods to construct a nonparametric frontier so that the efficiency relative to the frontier can be calculated and measured. The principle is to use the linear programming method and duality theorem to obtain the production frontier of each decision-making unit to calculate its relative efficiency. The model is divided into input- and output-oriented models. The analysis goal of this article is to determine the impact of blockchain on a company's innovation efficiency. Therefore, the inefficiency of the decision-making unit must be evaluated from the perspective of factor input, so the input-oriented model is selected:

$$\min\theta - \varepsilon(e^T S^m + e^T S^t)$$

$$\text{s.t.} \begin{cases} \sum_{j=1}^{n} X_j \lambda_j + S^- = \theta X_0 \\ \sum_{J=1}^{n} Y_j \lambda_j - S^+ = Y_0 \\ \lambda_j \geq 0, \qquad S^m, S^t \geq 0 \end{cases} \tag{1}$$

In Eq (1), there are $m$ types of input and $s$ types of output, which are represented by $X_j$ and $Y_j$, respectively. $X_{ij} > 0$ represents the i-th type of input of the j-th decision-making unit, and $Y_{rj}$ represents the r-th type of output of the j-th decision-making unit.

Considering the variable returns to scale in the company's actual production, the convexity constraint ($\sum_{i=1} \lambda i = 1, \lambda i \geq 0$) is introduced, and the input-oriented BCC model is selected to allow for variable returns to scale. For any decision-making unit, the BCC model divides each decision-making unit into technical efficiency (TE), pure technical efficiency (PTE), and scale efficiency (SE). Among them, comprehensive technical efficiency refers to the ability to achieve the minimum input under a given output, scale efficiency refers to the degree of economies of scale compared with the effective point of scale, and pure technical efficiency refers to the efficiency of excluding scale factors. The relationship between the three is TE = SE×PTE.

If $\theta = 1$, $s^+ = 0$, or $s^- = 0$, then the DEA of the decision-making unit is effective, and the slack variable is zero, which means that the economic activities of the decision-making unit are both technically efficient and effective; if $\theta = 1$, $s^+ \neq$ or $s^- \neq 0$, then the decision-making unit is weakly DEA effective, and its economic activity has one item that fails to achieve efficient production; if $\theta < 1$, then the decision-making unit is not DEA effective, and technical efficiency and scale efficiency cannot be simultaneously met. Therefore, the performance of the decision-making

unit is affected by management inefficiency, environmental factors and statistical noise, and it is necessary to separate these three effects through stochastic frontier analysis (SFA).

In the second stage, the SFA model is used to eliminate the influence of environmental factors and random interference on technical efficiency. In the stochastic frontier production function, the environmental variable is used as the explanatory variable, the slack variable in the first stage is regressed, and the selection is based on our first-stage orientation type. The first stage is input oriented, so SFA is performed only on the input slack variable regression decomposition to adjust all decision-making units to the same environment while considering the interference of random factors to adjust the input variables.

$$S_{ik} = f^i(z_k; \beta^i) + v_{ik} + \mu_{ik};$$
$$i = 1, 2 \ldots, m; k = 1, 2 \ldots, n \tag{2}$$

In Eq (2), $S_{ik}$ is the relaxation value of the i-th input of the k-th decision-making unit; $z_k$ is the environmental variable; $\beta^i$ is the coefficient of the environmental variable; $f^i(z_k; \beta^i)$ represents the influence of the environmental variable on the input slack $S_{ik}$; $v_{ik} + \mu_{ik}$ is the mixed error term; $V_{ik}$ is random interference, which refers to the influence of random disturbance factors on the input slack variable; and $\mu_{ik}$ measures management inefficiency, which refers to the influence of management factors on the input slack variable, assuming that it obeys the normal distribution truncated at the zero point, that is, $v \sim N(0, \sigma^2 v_i)$ and $\mu \sim N(0, \sigma^2 u_i)$. Moreover, $\gamma = \sigma^2 u_i / (\sigma^2 v_i + \sigma^2 u_i)$, $\gamma \sim (0,1)$. When $\gamma$ tends to 1, the influence of management factors is dominant in the inefficient decision-making unit; when $\gamma$ tends to 0, random error interference is dominant in inefficient decision-making unit. At the same time, the one-sided likelihood test statistic *LR* using the $\gamma$ null hypothesis can be used to test the rationality of the SFA model setting.

In the final stage, the data are adjusted through the coefficient estimation of the SFA regression, the decision-making unit is placed under the same environment and random interference to analyze the actual input, adjusted input value $x_{ij}$ and the original output value $y_{rj}$ obtained in the second stage are added to the DEA model again. The efficiency value obtained at this time is that after eliminating the influence of the environment and random errors, which is a truer and more accurate measure.

## Introduction to the superefficiency DEA model

The effective decision-making units in the classic DEA model cannot be distinguished according to their efficiency value, and there is no difference in the calculation results. However, in reality, there are differences between the effective decision-making units of the DEA, which cannot be reflected by the classic model. In 1993, Andersen and Petersen proposed a superefficiency DEA model to solve the efficiency value ranking of DEA-effective decision-making units. The basic idea is that when evaluating each decision-making unit of j0, it is removed from the constraints so that the DEA optimal value of the effective decision-making unit under the original model is greater than 1 according to the superefficiency DEA model. However, the optimal value of the non-DEA-effective decision-making unit from the original model remains unchanged under the superefficiency DEA model. Therefore, the problem of effective decision-making unit sequencing is solved.

## Data source and selection

Blockchain technology has been used since 2008 due to the emergence of cryptocurrency, while its practical applications in production, operation, and management have only begun to

**Table 2. Evaluation index of innovation efficiency based on DEA.**

| First-level index | Second-level index | Third-level index |
|---|---|---|
| Innovation inputs | Innovation expenditure | R&D expenditure as a proportion of main business income |
|  | Innovation labor capital | Proportion of R&D personnel among total employees |
| Innovation outputs | Intellectual property creation | Patent value |
|  | Innovation value realization | Investment income |

appear in the past 10 years. Due to the limited application cases of blockchain technology, there is a serious shortage of data for 2017 and 2018, resulting in a small sample size, which is not representative. Therefore, this article mainly evaluates the innovation efficiency of listed companies in 2019 and selects a total of 50 listed companies around the world, including 15 companies that apply blockchain technology and 35 companies that do not.

**Input indicators.**   When measuring enterprise innovation indicators, the related indicators—R&D expenditures and R&D personnel—are usually used as standards (e.g., Becheikh et al. [43]; Hu, [44]; Duran et al. [45]). This article uses the proportion of R&D expenditure among main business income and the proportion of R&D personnel among total employees as the input indicators to measure the impact of blockchain technology, as shown in Table 2.

**Output indicators.**   Combined with the existing research (e.g., Dewangan and Godse [46]; Fischer and Leidinger [47]) and the actual aim of this paper, investment income and patent value are used as output indicators, which can effectively provide feedback concerning the overall efficiency of listed companies, as indicated in Table 2.

To ensure the effectiveness and efficiency of DEA, the number of decision-making units is more than 3 times the number of input and output indexes in total. In this paper, there are 50 decision-making units and 4 input and output indexes that meet the DEA application standards. In addition, the correlation between input and output indicators needs to be tested in advance. Because the data do not conform to a normal distribution, according to Spearman's nonparametric test calculation, there is a significant positive correlation between input and output indicators. The correlation coefficients between investment income and the proportion of R&D expenditure and investment income and the proportion of R&D personnel are 0.54 and 0.6, respectively, at 1% and 5% significance, respectively. The correlation coefficients between patent value and the proportion of R&D expenditure and patent value and the proportion of R&D personnel are 0.64 and 0.72, respectively, at 5% significance. This finding shows that the index selection in this paper is reasonable and conforms to the assumption of the "same direction" of input and output indexes. This index system can be used to analyze innovation efficiency.

**Indicators of environmental effects.**   Regarding the selection of indicators for influencing factors, when conducting SFA modeling, environmental variables should satisfy the "separation assumption", that is, to select those factors that have an impact on the innovation efficiency of listed companies but beyond the subjective control of the sample. According to existing research (e.g., Monroe-White and Zook [48]; Piperopoulos et al. [49]; Rampersad and Troshani [50]; Wang et al. [51]) and given the attributes and development characteristics of listed companies, the macroeconomic level, the degree of opening to the outside world, the level of informatization, and the degree of regional technological development are considered external uncontrollable factors and are thus used as environmental variables.

- Macroeconomic level. Enterprise innovation activities are inseparable from labor and capital investment, and the degree of local economic development brings about diversified resource endowments, which in turn affect the market performance of listed companies. In this

article, the per capita GDP of the province (China)/state (U.S., Europe) where the listed company is located is used to measure the level of macro development.

- Degree of opening to the outside world. A high degree of regional openness to the outside world leads to a rapid flow of production factors, promotes technological exchanges and information exchanges, enhances the competitiveness of enterprises, and increases the innovation power of enterprises and the multiplier effect brought about by accelerated innovation, which directly affects the market performance of enterprises. In this article, the proportion of foreign direct investment (FDI) in GDP is used to measure the degree of openness of the region where the listed company is located.

- Informatization level. The application of new technologies, especially the enormous information support required by blockchain, is inseparable from the construction of the information foundation. This article uses the Internet broadband access user data of the province (China)/state (U.S., Europe) where the listed company is located to measure the degree of local informatization development.

- Degree of regional technological development. The ranking of global technology clusters can be obtained from the annual "Global Innovation Index" report of the World Intellectual Property Organization. The regional scientific and technological development of listed companies is measured according to the proportion of the number of applications and scientific publications filed by the cluster of listed companies based on the "Patent Cooperation Treaty" (PCT) in the global total.

In the selection of environmental data, due to the different publication standards of macro data in different regions and considering the scope of technology dissemination, the macro data of the province (China)/state (U.S., Europe) where the listed company is located is selected as a reference. For a small number of regions without direct data on technology clusters, the clusters of the nearest neighboring cities are used as a data reference (technology spillover effect). This paper uses the three-stage DEA model and the superefficiency DEA model to evaluate the relative efficiency and compare the market performance of listed companies' economic activities.

## Results

### First stage: Empirical analysis with traditional DEA model

In the first phase, DEAP2.1 is used to analyze the efficiency of the cultural input and output of 50 listed companies in 2019, as shown in Table 3.

The results show that the average comprehensive technical efficiency of the 50 listed companies is 0.396 in 2019, the average pure technical efficiency is 0.627, and the average value of scale efficiency is 0.578. Among the inefficient decision-making units, the pure technical efficiency of 25 listed companies is significantly greater than their scale efficiency. It can be seen from the returns to scale data that except for the constant returns to scale of listed companies in the frontier of efficiency and the diminishing returns to scale of Meicai Technology, the rest of the listed companies are in the stage of increasing returns to scale, which means that scale efficiency can be achieved by expanding the innovation scale, thus achieving the best overall efficiency. There are two companies at the frontier of efficiency, namely, Amazon and Yatai Group, accounting for 4% of the total number of companies, both of which are companies that use blockchain technology, accounting for 13.33% of the number of companies applying blockchain technology for production. Among the companies that have introduced blockchain technology into economic activities, three have achieved the best pure technical efficiency

**Table 3. DEA efficiency value in the first stage.**

| Company | TE | PTE | SE | RS | company | TE | PTE | SE | RS |
|---|---|---|---|---|---|---|---|---|---|
| Kaisa Health | 0.806 | 0.975 | 0.827 | irs | Beihua | 0.165 | 0.368 | 0.448 | irs |
| Amazon | 1 | 1 | 1 | - | Shaanxi Natural Gas | 0.406 | 0.883 | 0.459 | irs |
| DRAFTKINGS | 0.825 | 0.989 | 0.834 | irs | Hongtao Shares | 0.228 | 0.505 | 0.452 | irs |
| EMERALD | 0.774 | 1 | 0.774 | irs | *ST Chenxin | 0.114 | 0.255 | 0.447 | irs |
| MGT Capital Investment | 0.415 | 0.512 | 0.81 | irs | Oriental Tower | 0.24 | 0.523 | 0.46 | irs |
| Sohu | 0.143 | 0.957 | 0.15 | irs | Alto Electronics | 0.109 | 0.243 | 0.447 | irs |
| 500 Lottery Network | 0.661 | 0.804 | 0.822 | irs | Yao Ji Technology | 0.115 | 0.259 | 0.442 | irs |
| Guangbo Shares | 0.827 | 0.999 | 0.828 | irs | Kingway | 0.241 | 0.538 | 0.448 | irs |
| Xinlong Health | 0.822 | 0.995 | 0.825 | irs | Unilumin Technology | 0.156 | 0.351 | 0.445 | irs |
| Lisheng Racing | 0.821 | 0.992 | 0.828 | irs | Enlightenment Design | 0.138 | 0.305 | 0.452 | irs |
| Lehman Optoelectronics | 0.82 | 0.991 | 0.828 | irs | Xuanya International | 0.093 | 0.209 | 0.443 | irs |
| China Sports Industry | 0.845 | 1 | 0.845 | irs | Nanjing Julong | 0.166 | 0.371 | 0.447 | irs |
| *ST Union | 0.827 | 0.999 | 0.828 | irs | Jinling Sports | 0.163 | 0.365 | 0.446 | irs |
| Zhejiang Shu Culture | 0.801 | 0.965 | 0.83 | irs | Chuangyuan Culture | 0.117 | 0.261 | 0.447 | irs |
| Yatai Group | 1 | 1 | 1 | - | Shanghai Construction | 0.212 | 0.432 | 0.49 | irs |
| Zhongshi Jinqiao | 0.448 | 1 | 0.448 | irs | *ST Han Ye | 0.144 | 0.264 | 0.545 | irs |
| 361 degrees | 0.242 | 0.52 | 0.466 | irs | Guanhao High-tech | 0.119 | 0.268 | 0.445 | irs |
| Xtep International | 0.224 | 0.499 | 0.449 | irs | Seiko Steel | 0.171 | 0.367 | 0.466 | irs |
| Anta Sports | 0.15 | 0.816 | 0.183 | irs | Jihua Group | 0.271 | 0.638 | 0.424 | irs |
| Li Ning | 0.405 | 0.647 | 0.626 | irs | Aokang International | 0.262 | 0.586 | 0.448 | irs |
| EVENTBRITE | 0.482 | 1 | 0.482 | irs | Dafeng Industry | 0.172 | 0.385 | 0.447 | irs |
| Huya Live | 0.224 | 0.482 | 0.465 | irs | Center Shares | 0.169 | 0.381 | 0.444 | irs |
| Nautilus Sports | 0.363 | 0.813 | 0.447 | irs | *ST Noble | 0.191 | 0.42 | 0.454 | irs |
| Meicai Technology | 0.95 | 1 | 0.95 | drs | Kolida | 0.189 | 0.422 | 0.447 | irs |
| Sirius XM | 0.41 | 0.41 | 1 | - | Baiyun Electric | 0.17 | 0.375 | 0.453 | irs |
| **Mean** | **0.396** | **0.627** | **0.578** | | | | | | |

Note: RS stands for short returns to scale, ins stands for increased returns to scale, drs stands for diminishing returns to scale, and—stands for constant returns to scale. The shaded part in Table 3 represents the 15 listed companies that have applied blockchain.

values, namely, Zhongshi Jinqiao, EVENBRITE, and Meicai Technology; one listed company, Sirius XM, has achieved the best scale efficiency. The above companies have shown higher innovation efficiency than the other companies, accounting for 11.43%. The remaining companies thus have room for improvement in terms of scale efficiency and pure technical efficiency.

To further explore the impact of blockchain technology on companies' innovation efficiency, this article compares the listed companies by splitting them into two groups, as displayed in Table 4.

According to the statistics in Table 4, the average performance of each efficiency value of the listed companies that use blockchain technology to participate in economic activities is better than that those that do not use blockchain technology. Independent sample testing finds that the p values of overall technical efficiency and scale efficiency are less than 0.05, rejecting H0 that there is no difference in the mean of the two samples. Moreover, the difference in the mean of the data efficiency of the two groups is statistically significant, and the pure technical efficiency p value is less than 0.05, rejecting H0 that the mean of the two samples is not different. The difference in the mean is statistically significant. This independent sample t test significantly compares whether there are companies that use blockchain technology, indicating that

**Table 4. Grouping statistics.**

| Grouping statistics | | Number of cases | Mean | T | Sig. |
|---|---|---|---|---|---|
| Crste | Blockchain-applied | 15 | 0.759133 | 8.897 | 0.000*** |
| | Nonblockchain | 35 | 0.240543 | | |
| Vrste | Blockchain-applied | 15 | 0.945200 | 6.716 | 0.000**** |
| | Nonblockchain | 35 | 0.490314 | | |
| Scale | Blockchain-applied | 15 | 0.801933 | 9.369 | 0.000**** |
| | Nonblockchain | 35 | 0.481771 | | |

***p<0.01

**p<0.05

and

*p<0.1.

blockchain technology has a significant supporting effect on economic activities, and companies that have not reached the best pure technical efficiency can optimize innovation efficiency by introducing blockchain technology for promotion.

## Second stage: Empirical analysis of SFA regression

The slack of the input variables of the decision-making unit obtained above are used as dependent variables, and the environmental variables are used as independent variables to carry out SFA regression. The results are shown in Table 5 through the use of Frontier4.1. It can be seen from Table 5 that in the two models with input slack variables, all influencing factors, except users with broadband Internet access, pass the 1% or 5% significance test, indicating that the selection of the variables in the model is reasonable.

The input slack variable refers to the amount of input that may be reduced by improving the management level. Therefore, if the environmental variable is positively correlated with the input slack variable, then it means that an increase in environmental variable input will not be conducive to the improvement of efficiency. From the regression analysis in the above table, it can be concluded that the LR of the regression model of the two input indicators reaches a 1% significance level, indicating that the environmental variables selected in this paper are reasonable. The γ value approaches 1, indicating that the slack of R&D investment is caused by management inefficiency. When the regression coefficient is negative, the increase

**Table 5. Results of SFA regression.**

| Environmental factor (Independent variable) | | Slack of R&D expenditure proportion | Slack of R&D personnel proportion |
|---|---|---|---|
| | | coefficient | coefficient |
| Constant | 0 | -140.91 *** | -36.38 *** |
| Per capita GDP | 1 | -1.25 ** | -2.28** |
| Proportion of FDI in GDP | 2 | -670.02*** | -405.94*** |
| Users with broadband Internet access | 3 | 0.02 ** | 0.05 |
| Science and technology cluster index | 4 | -12.87 *** | -5.93 ** |
| $\sigma^2$ | | 60,498.65*** | 130,941.54*** |
| γ | | 1*** | 1*** |
| Log likelihood function | | -319.40 | -333.63 |
| LR test of the one-sided error | | 15.36*** | 17.33*** |

Note: *** and ** represent significance levels of 1% and 5%, respectively.

in environmental variables helps reduce the amount of input slack; in contrast, this means that the increase in the values of the environmental variables has a positive correlation with the increase in the values of the input slack variables. According to the regression coefficients of the four environmental variables on the slack of each input, the following can be seen:

- Per capita GDP measures the level of local economic development. The per capita GDP of the company's location has a significant negative effect on the slack of R&D expenditure proportion and the slack of R&D personnel proportion, indicating that the improvement of economic development has brought about a good market environment, which promotes healthy competition and reduces both R&D expenditure and R&D personnel consumption. The wasting of personnel input urges enterprises to lower their costs and improve their innovation and operations efficiency.

- FDI, that is, foreign direct investment, as a proportion of GDP, is significantly negatively correlated with the slack of R&D expenditure proportion and the slack of R&D personnel proportion. An increase in the proportion of FDI is conducive to saving R&D funds and the input factors of R&D employees. The openness of the region not only introduces cutting-edge R&D technology and management systems but also accelerates the flow and reasonable distribution of regional personnel, which is conducive to information exchange and innovation efficiency for enterprises, leading to a more significant improvement.

- Users with broadband Internet access have a weaker impact on the R&D expenditure proportion and the proportion of R&D personnel investment variable slack and are positively correlated, which means that they bring about the wasting of R&D funds and R&D personnel, but fortunately, this impact is minimal.

- The negative impact of the science and technology cluster index on the slack of R&D expenditures proportion is greater than that on the slack of R&D personnel proportion, denoting that regional technological innovation has a spillover effect, accelerating the technological upgrading of surrounding enterprises; regional technological spillover not only reduces the wasting of R&D personnel but also contributes more to reducing the investment of R&D funds and realizing the improvement of enterprise efficiency at a lower cost.

In summary, the input factors that affect innovation efficiency are all affected by environmental factors. Among them, the economic level, openness and cluster innovation index all show significant effects. For the sake of scientific and true measurement results, 50 sample companies are analyzed under identical environmental conditions and random interference by isolating the impacts from managerial performance and environmental effects. Therefore, on the basis of the SFA regression results, the original input data are adjusted to eliminate environmental interference, and optimized data can be obtained.

### Third stage: Empirical analysis of adjusted DEA

The use of DEAP2.1 once again facilitates the analysis of the adjusted input data of the BCC-DEA model and accurately measures the innovation efficiency of 50 listed companies, as shown in Table 6.

**Analysis of the overall innovation efficiency of blockchain technology.** By comparing the DEA efficiency values of the first stage with those of the third stage, an additional 4 companies that have reached the frontier of efficiency, namely, EMERALD, Xinlong Health, Lehman Optoelectronics and Meicai Technology, are identified, which shows that a total of six listed companies are effective not only in terms of the technical efficiency of their innovation investment but also in terms of their scale efficiency. The average comprehensive efficiency of the 50

**Table 6. DEA efficiency value in the Third Stage (Adjusted input data).**

| Company | TE | PTE | SE | | Company | TE | PTE | SE | |
|---|---|---|---|---|---|---|---|---|---|
| Kaisa Health | 0.528 | 0.552 | 0.957 | drs | Beihua | 0.306 | 0.569 | 0.538 | irs |
| Amazon | 1 | 1 | 1 | - | Shaanxi Natural Gas | 0.366 | 0.617 | 0.594 | irs |
| DRAFTKINGS | 0.87 | 0.886 | 0.981 | drs | Hongtao Shares | 0.337 | 0.556 | 0.607 | irs |
| EMERALD | 1 | 1 | 1 | - | *ST Chenxin | 0.155 | 0.257 | 0.602 | irs |
| MGT Capital Investment | 0.48 | 0.554 | 0.867 | irs | Oriental Tower | 0.283 | 0.454 | 0.622 | irs |
| Sohu | 0.143 | 0.677 | 0.211 | irs | Alto Electronics | 0.182 | 0.301 | 0.604 | irs |
| 500 Lottery Network | 0.957 | 1 | 0.957 | drs | Yao Ji Technology | 0.16 | 0.267 | 0.598 | irs |
| Guangbo Shares | 0.869 | 0.887 | 0.979 | drs | Kingway | 0.276 | 0.46 | 0.6 | irs |
| Xinlong Health | 1 | 1 | 1 | - | Unilumin Technology | 0.25 | 0.413 | 0.605 | irs |
| Lisheng Racing | 0.653 | 0.658 | 0.993 | drs | Enlightenment Design | 0.2 | 0.323 | 0.62 | irs |
| Lehman Optoelectronics | 1 | 1 | 1 | - | Xuanya International | 0.146 | 0.242 | 0.605 | irs |
| China Sports Industry | 0.856 | 0.921 | 0.929 | drs | Nanjing Julong | 0.21 | 0.347 | 0.604 | irs |
| *ST Union | 0.574 | 0.577 | 0.995 | drs | Jinling Sports | 0.292 | 0.533 | 0.547 | irs |
| Zhejiang Shu Culture | 0.716 | 0.719 | 0.996 | drs | Chuangyuan Culture | 0.195 | 0.338 | 0.576 | irs |
| Yatai Group | 1 | 1 | 1 | - | Shanghai Construction | 0.25 | 0.379 | 0.66 | irs |
| Zhongshi Jinqiao | 0.355 | 0.586 | 0.606 | irs | *ST Han Ye | 0.218 | 0.285 | 0.766 | irs |
| 361 Degrees | 0.4 | 0.747 | 0.535 | irs | Guanhao High-tech | 0.235 | 0.437 | 0.536 | irs |
| Xtep International | 0.314 | 0.558 | 0.563 | irs | Seiko Steel | 0.213 | 0.333 | 0.639 | irs |
| Anta Sports | 0.238 | 1 | 0.238 | irs | Jihua Group | 0.31 | 0.541 | 0.573 | irs |
| Li Ning | 0.456 | 0.606 | 0.753 | irs | Aokang International | 0.45 | 0.84 | 0.536 | irs |
| EVENTBRITE | 0.513 | 0.912 | 0.563 | irs | Dafeng Industry | 0.251 | 0.431 | 0.583 | irs |
| Huya Live | 0.517 | 0.969 | 0.534 | irs | Center Shares | 0.23 | 0.384 | 0.598 | irs |
| Nautilus Sports | 0.3 | 0.579 | 0.518 | irs | *ST Noble | 0.305 | 0.552 | 0.553 | irs |
| Meicai Technology | 1 | 1 | 1 | - | Kolida | 0.246 | 0.409 | 0.601 | irs |
| Sirius XM | 0.698 | 0.698 | 1 | - | Baiyun Electric | 0.251 | 0.427 | 0.587 | irs |
| mean | 0.455 | 0.616 | 0.703 | | | | | | |

Note: The shaded part in Table 6 represents the 15 listed companies that have applied blockchain.

listed companies increases from 0.396 to 0.455; the average pure technical efficiency value decreases from 0.627 before adjusting the input data to 0.616. Although this change is small, the impact of environmental factors shows that the actual management level of the 50 sample companies is overrated; the average scale efficiency increases significantly from 0.578 to 0.703 after adjusting the input data. At the same time, the emergence of 8 listed companies that are in the stage of diminishing returns to scale in production exhibits the clumsiness of organizations. Moreover, these 8 companies have not applied blockchain technology. Except for the 6 companies with constant returns to scale, the remaining 44 listed companies are in the stage of increasing returns to scale, indicating that most companies' innovation inputs have not reached the maximum optimal scale, which restricts the improvement of enterprise innovation efficiency. Insufficient actual innovation efficiency is mainly caused by the low efficiency of pure technology.

## Analysis of specific indicators

- After adjusting the input data, the overall efficiency of a total of 40 listed companies was improved. The largest increase was for Huya Live, which was 130.8%, and the overall efficiency of 7 companies decreased. The company with the largest decrease was Kaisa Health,

which did not exceed 35%. The average overall efficiency value of the 50 listed companies increased by 30.6%, while the number of companies reaching the innovation frontier increased from 2 to 6. Newly added companies included EMERALD, Xinlong Health, Lehman Optoelectronics and Meicai Technology, of which the top three are listed companies that have invested in blockchain technology. After the above adjustment, a total of 5 blockchain technology application companies were found to have reached frontier efficiency.

- After the adjustment of investment in pure technical efficiency, 26 companies showed growth, 4 companies remained unchanged, and the remaining 20 companies showed a decrease in pure technical efficiency, 8 of which were from the applied blockchain technology group and 12 of which were from the nonblockchain technology group. Among them, the largest increase was shown for Huya Live, reaching 101%, and the largest decrease came from Kaisa Health, dropping 43.4%.

- After the adjustment of scale efficiency, except for Amazon, Yatai Group and Sirius XM, the remaining 47 companies improved across the board, with an average increase of 26.1%. The average increase in the scale efficiency of listed companies that have applied blockchain technology was 21.4%, and the average increase in the scale efficiency of listed companies that have not was 27.8%.

- Comparing pure technical efficiency and scale efficiency, it can be seen from the DEA model that they jointly determine the comprehensive innovation efficiency of listed companies. From the analysis results, scale efficiency is found to perform better than pure technical efficiency. A total of 34 of the 44 listed companies had greater scale efficiency than pure technical efficiency. Only 10 companies, Sohu, 500 Lottery, 361 degrees, Anta Sports, EVENTBRITE, Huya Live, Nautilus Sports, Beihua, Shaanxi Natural Gas, and Aokang International, had better performance in terms of pure technology efficiency. The largest difference between pure technology efficiency and scale efficiency was found for Anta Sports, at 0.762; its pure technical efficiency reached the optimal value, but its scale efficiency was seriously insufficient.

## Superefficiency DEA model results

According to the analysis of the three-stage DEA model, after the adjustment of input data, the superior companies at the efficiency frontier are Amazon, EMERALD, Xinlong Health, Lehman Optoelectronics, Yatai Group, and Meicai Technology. In Table 7, the above six companies are sorted by the superefficiency DEA model to provide further effective distinctions among them, and the results obtained after using MATLAB for calculation are presented.

The superefficiency scores in descending order are those for Amazon, Yatai Group, EMERALD, Xinlong Health, Meicai Technology, and Lehman Optoelectronics, with scores of 3.3395, 1.3309, 1.2057, 1.0231, 1.0089, and 1, respectively. According to the meaning of superefficiency analysis, Amazon is still at the efficiency frontier after the initial investment has increased by 233.95%. Similarly, except for Lehman Optoelectronics, several other companies

**Table 7. Empirical results from the superefficiency DEA model.**

| Amazon | EMERALD | XinLong Health | Lehman Optoelectronics | Yatai Group | Meicai Technology |
|--------|---------|----------------|------------------------|-------------|-------------------|
| 3.3395 | 1.2057 | 1.0231 | 1.0000 | 1.3309 | 1.0089 |

Note: Listed companies that perform insufficiently according to the efficiency values obtained through the superefficiency DEA model, stay unchanged as before and thus are not argued here.

can still be effective in terms of innovation when the original investment increases by a certain percentage: 33.09%, 20.57%, 2.31% and 0.89%. Before data adjustments were made, only two companies that applied blockchain technology, Amazon and Asia Pacific Group, were at the efficiency frontier, and they were still in the top two positions after the adjustment. Moreover, after the adjustment, 3 of the 4 companies that were added were all from the application of the blockchain group. Five blockchain innovation frontier companies accounted for 33.33% of the same group, while nonblockchain application companies only achieved effective innovation at a level of 2.86% for the group. This finding reveals that in the sample grouping comparison, the overall performance of blockchain-applying companies is better than that of nonblock-chain-applying companies, their innovation efficiency is higher, and there are more companies innovating effectively.

## Discussion

After the analysis of the three-stage DEA and superefficiency DEA models, the innovation efficiency of companies with blockchain is found to be generally higher than that of companies without blockchain among the 50 listed sports companies, which means that the application of blockchain technology or concepts can improve innovation efficiency. In other words, the problem of low innovation efficiency caused by the current business model of the sports industry can be solved by blockchain. Therefore, through the above empirical analysis, we have explained the necessity of business model innovation in the sports industry based on blockchain technology, and the internal logic and structure of its business model innovation is illustrated with the case of *vSport*.

The global sports industry market is enormous and has diverse profit models. Athlete training and education, ticket sales, broadcast rights sales, commercial advertising sponsorship, derivative product development, sports lotteries and user data monetization all are included in the sports industry. These seemingly related parties are in a state of segmentation. On the one hand, due to the intervention and control of the central organization, the effective connection of upstream and downstream resources in the sports industry is hindered; on the other hand, affected by the geographical and market development environment, there is a general problem in the global sports industry—the influence is far greater than the profit, and "only cheering but not money" is the embarrassing status quo. For example, most of the income of the head IP is collected by several lead agents and related intermediaries, and the sports stars themselves do not have the right to speak or obtain much income. Although sports stars generally deal with brokers or brokerage companies, most of them operate through traditional intermediary channels, among which there are too many middlepeople, such as Mendes predators. However, the emergence of blockchain technology can completely change this situation.

Therefore, the business model of the blockchain-based sports industry realizes mutual conductance between enterprises and operators in terms of traffic, products and services; reduces transaction costs; and can further stimulate consumers' consumption potential (see Fig 5).

As shown in the above figure, the upper part is the application layer, which builds a distributed business system. Enterprises and consumers realize the exchange and mutual conduction between traffic and products/services based on the blockchain, which reduces transaction costs. The lower part is the middle-ground service layer, which improves the monetization ability of operators. The data flow, capital flow, and fan flow generated by the distributed business form finally form big data, digital assets and traffic value added. Enterprise operators can use big data intelligent analysis to achieve precision marketing, and small merchants can enjoy the dividends brought about by big data. Through the realization of data and traffic, the value added of digital assets is finally achieved.

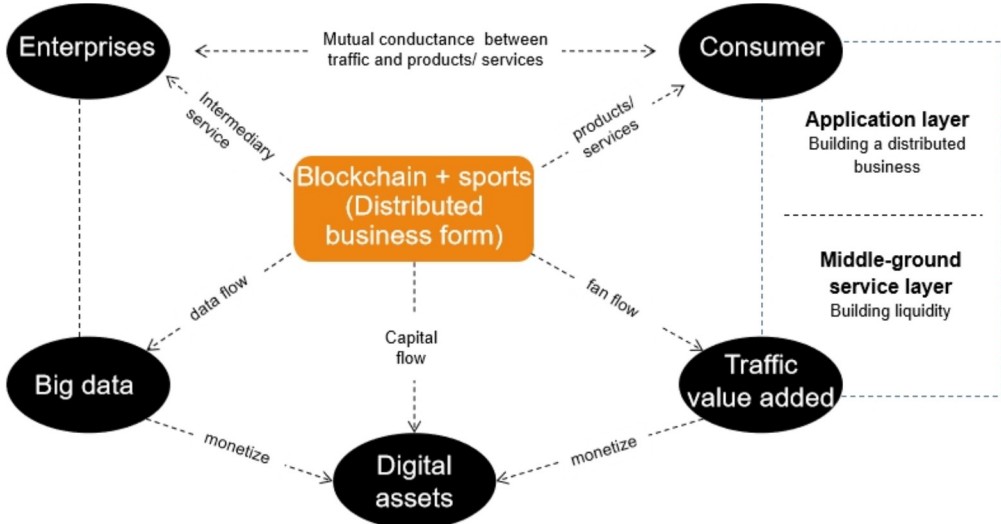

**Fig 5. The internal logic diagram of the business model of the sports industry based on blockchain.**

The following section takes the sports blockchain *vSport* as an example to specifically analyze whether the business model of the blockchain-based sports industry is feasible.

*vSport*, created by the Sports Value Foundation, is committed to creating a public chain serving the global sports industry. It is the first fully open and nonprofit blockchain platform for the global sports industry. At the end of 2017, Qiang Bai and Dutch football superstar Sneijder jointly established vS*port*. By issuing VSC tokens, they can express, connect, and better circulate value. It is a better way to monetize the value of sports. At present, its tokens have been listed on two digital asset exchanges: Manbi and bcex.

## Financing and asset appreciation

Many sports companies are faced with financing difficulties and long payback periods, such as building stadiums, which require cash flow, but the cash payback period is relatively long. For example, to build a stadium, even if the money needed is not at an extremely high value, it will take more than 5 years to pay back the cost. Since equity financing has an exit period of 5+1 years, the capital payback period for stadium construction lasts for 5 years, which means that before making a profit, the capital takes dividends and withdraws them; thus, issuing tokens is a possible solution. At this time, the term "token" here is equivalent to the role of equity, representative of securitization, but it is different from equity. It is a more flexible financing method that needs to carried out for a long time and is not a currency for circulation. After holding it for a long time, for example, if the ice and snow fields and football fields earn income, then dividends can be distributed, and the token can also be sold on the exchange after it has been revalued.

Both equity and the token must be measurable. For example, for browsers, they are measured by the number of users and must be anchored to something in reality. What is the anchor value of the token among sports companies? It is certainly not simply a combination of profit and income. *vSport* is practicing to integrate various types of sports activities, such as clubs, players, associations, etc., and to create a sports community. As long as the various business formats of the real world are gradually combined with tokens, we can know how to combine the price of cryptocurrency with the actual value. *vSport* negotiates with various actors in the sports industry to discover the rigid needs of each in reality and finally combines these rigid needs with cryptocurrency.

## Value added and liquidity of traffic

In the Internet age, traffic has become a very important asset. Many sports companies have traffic, such as forum communities, offline football, and fitness industries. They all have traffic, but how to monetize this traffic remains difficult. *vSport* has signed 433 athletes, including Ronaldo, Neymar and other active players, with more than 26 million fans around the world and is the world's largest football social media network, and thus, it is difficult to monetize. Even the promotion of related sports products on *vSport* cannot change the habit of people using e-commerce to buy goods, so it is difficult to find an appropriate monetization model. Therefore, *vSport* is trying to convert the fans of these athletes into users of its currency circle so that they can profit from blockchain tokens.

## The benefits of scale brought about by distributed business systems

Many sports companies are doing well in certain areas, but it is difficult to replicate their approaches because sports are inherently fragmented. For example, American football is the largest participation sport in the United States, but there is no dominant company in the football training industry because it is difficult to scale. If using a token and then measuring it according to the workload of all parties, for example, to issue tokens to each different training institution and establish a benefit distribution mechanism, they become integrated. This is what the blockchain brings—the transformation of production relations and organizational forms.

In this way, we can combine sports companies in different regions to form a new organization. The corporate behaviors of all parties can create value for such an organization so that its overall income increases, and then, the benefits are distributed according to the workload recorded by blockchain, so the weight of negotiations with sponsors also increases. There are more than 350 youth football clubs under *vSport*, none of which has more than 1,500 people, which makes it difficult to negotiate terms with sponsors. For example, Nike may not invite Ronaldo to a club with 1,500 people, but there are more than 350 clubs, each covering 1,000 families, so Nike would have enough motivation to do so. The original problem is that how to quantify the contribution of each club to distribute the benefits fairly could be difficult. Now, this problem can be solved through blockchain technology. In short, the best use of tokens for the sports industry is to realize the influence of sports.

## Limitations

First, the sample companies studied in this article are listed. Since listed companies are more likely to pay more attention to innovation and to try out new technologies and concepts, it is necessary to use samples of other companies for further research to test the consistency of the results.

Second, in the selection of evaluation indicators for evaluating innovation efficiency, due to the objective limitation of data collection, there is no way to collect more relevant indicators and data, such as those on patent licensing and transfer revenue, new product sales revenue, and new product marketing revenue. Future research can collect more relevant data through field inspections, interviews, and surveys, making the necessity for sports companies to introduce blockchain more convincing.

Finally, although we have used the case of the sports blockchain *vSport* to illustrate the feasibility of this business model, the development of blockchain is still in its infancy, and there are very limited examples of the combination of blockchain and the sports industry. In the future, the combination of these two areas will become more extensive and in-depth, and this kind of business model innovation will also face new challenges and problems, which will require further investigation and research.

## Conclusions

From the perspective of necessity and possibility, this article provides empirical support and proposes the structure of business model innovation in the sports industry based on blockchain. First, the three-stage DEA efficiency model proves that listed sports companies that use blockchain technology or concepts have obvious advantages in terms of innovation efficiency compared with listed companies that have not adopted such technology. Second, through the analysis of existing business models of the sports industry, we propose a sports industry business model based on blockchain and its internal logic diagram. Finally, the analysis of the example of the sports blockchain *vSport* illustrates the possibility of the combination of sports and blockchain, providing new ideas and insights for the business model innovation of the sports industry.

## Supporting information

**S1 Table. All used data sets.** It contains the data used in the article.
(XLS)

**S2 Table. Summary statistics and correlations.** This table presents the means, standard deviations, and correlations. Most significant correlations (e.g., between investment income and proportion of R&D personnel in total employees, and between patent value and R&D expenditure as a proportion of main business income) are as expected.
(PDF)

**S1 File.**
(PDF)

## Author Contributions

**Conceptualization:** Chenchen Lv, Yifeng Wang.

**Data curation:** Chai Jin.

**Formal analysis:** Chenchen Lv.

**Investigation:** Chenchen Lv.

**Methodology:** Chai Jin.

**Project administration:** Yifeng Wang.

**Resources:** Yifeng Wang.

**Software:** Chai Jin.

**Supervision:** Yifeng Wang.

**Visualization:** Chenchen Lv.

**Writing – original draft:** Chenchen Lv.

**Writing – review & editing:** Chenchen Lv.

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
