## [Decision Letter · Decision Letter 0]

16 Dec 2021

The possibility of sports industry business model innovation based on Blockchain technology: Evaluation of the innovation efficiency of listed sports companies

PONE-D-21-21031

Dear Dr. LV,

We’re pleased to inform you that your manuscript has been judged scientifically suitable for publication and will be formally accepted for publication once it meets all outstanding technical requirements.

Kind regards,

Rashid Mehmood, PhD

Academic Editor

PLOS ONE

Reviewers' comments:

Reviewer's Responses to Questions

**Comments to the Author**

1. Is the manuscript technically sound, and do the data support the conclusions?

Reviewer #1: Yes

Reviewer #2: Yes

2. Has the statistical analysis been performed appropriately and rigorously? 

Reviewer #1: Yes

Reviewer #2: Yes

3. Have the authors made all data underlying the findings in their manuscript fully available?

Reviewer #1: No

Reviewer #2: Yes

4. Is the manuscript presented in an intelligible fashion and written in standard English?

Reviewer #1: Yes

Reviewer #2: Yes

5. Review Comments to the Author

Reviewer #1: Thank you for this exciting research that clarifies the influential role blockchain technology plays in business models for many areas, including the sports sector. The way of narration and dividing the search from general to specific was great.

Reviewer #2: This paper advocates the use of blockchain for business model innovation in the sports industry by evaluating the innovation efficiency of 50 sports companies that have and have not adopted the use of blockchain.

This paper is written well and offers good novelty and contributions. I recommend accepting it.

6. PLOS authors have the option to publish the peer review history of their article (what does this mean?). If published, this will include your full peer review and any attached files.

Reviewer #1: No

Reviewer #2: No

---

## [Editor Report · Acceptance letter]

4 Jan 2022

PONE-D-21-21031 

The possibility of sports industry business model innovation based on blockchain technology: Evaluation of the innovation efficiency of listed sports companies 

Dear Dr. LV:

I'm pleased to inform you that your manuscript has been deemed suitable for publication in PLOS ONE. Congratulations! Your manuscript is now with our production department. 

Kind regards, 

on behalf of

Dr. Rashid Mehmood 

Academic Editor

PLOS ONE